# An IoT System for Air Pollution Monitoring with Safe Data Transmission

**DOI:** 10.3390/s24020445

**Published:** 2024-01-11

**Authors:** Janusz Bobulski, Sabina Szymoniak, Kamila Pasternak

**Affiliations:** Department of Computer Science, Czestochowa University of Technology, 42-201 Czestochowa, Poland; sabina.szymoniak@icis.pcz.pl (S.S.); kamila.pasternak@pcz.pl (K.P.)

**Keywords:** IoT, wireless cloud system, air pollution environment protection

## Abstract

Air pollution has become a global issue due to rapid urbanization and industrialization. Bad air quality is Europe’s most significant environmental health risk, causing serious health problems. External air pollution is not the only issue; internal air pollution is just as severe and can also lead to adverse health outcomes. IoT is a practical approach for monitoring and publishing real-time air quality information. Numerous IoT-based air quality monitoring systems have been proposed using micro-sensors for data collection. These systems are designed for outdoor air quality monitoring. They use sensors to measure air quality parameters such as $CO_{2}$, CO, $PM_{10}$, $NO_{2}$, temperature, and humidity. The data are acquired with a set of sensors placed on an electric car. They are then sent to the server. Users can subscribe to the list and receive information about local pollution. This system allows real-time localized air quality monitoring and sending data to customers. The work additionally presents a secure data transmission protocol ensuring system security. This protocol provides system-wide attack resiliency and interception, which is what existing solutions do not offer.

## 1. Introduction

This paper is an extended version of our conference paper: “Air pollution monitoring and information distribution system”, Proceedings of the CORES and IP&C Conferences 2023 [1].

Rapid urbanisation and industrialisation have given rise to a global issue of air pollution. Exceeding recommended national limits is a common issue faced by many countries. Moreover, it should be emphasised that there are studies that suggest no level of exposure to single pollutants below which adverse health effects are not seen [2]. Bad air quality’s most significant environmental health risk significantly impacts Europe’s population. Bad air quality is known to cause various health problems, like lower respiratory infections, trachea, bronchus, lung cancers, ischemic heart diseases and strokes. Air pollutants can be divided into two categories: internal (indoor) and external (outdoor) air pollutants [3]. Air pollution that affects the entire atmosphere is called outdoor air pollution. This pollution type is caused by various air pollutants emitted into the atmosphere from sources such as automobiles, chemical industries and the burning of fossil fuels. Such pollutants mainly consist of carbon monoxide CO, ozone $O_{3}$, nitrogen oxides $NO_{x}$, sulfur dioxide $SO_{2}$, and particulate matter of diverse particle measures [4]. According to the WHO [5], the number of ambient air pollution-attributable deaths in 2019 exceeded 4,100,000 worldwide, 500,000 in Europe and 36,000 in Poland. It should be emphasised that internal air pollution is as severe a problem as outdoor air pollution. Air pollution classified as indoor can be found in buses, metro, offices, hospitals, schools, libraries, etc. [6]. One common source of internal air pollution is using polluting fuels and devices in households. Significant indoor air impurities are $NO_{x},\;SO_{2},\;O_{3},\;CO$, PM and microorganisms. Exposure to indoor air pollutants can result in harmful health outcomes. Household air pollution deaths 2019 exceeded 3,200,000 worldwide and 150,000 in Europe [7].

Since air pollution is a critical global issue, creating a system for monitoring and publishing real-time air-grade knowledge seems reasonable. The Internet-of-Things (IoT) is an approach that can be a suitable and effective solution to this issue. The IoT is a network that collects vast doses of data from diverse devices connected to more extensive schemes. The data gathered by this network are converted into useful information. More detailed knowledge about IoT can be found in [8].

The main contributions of this paper are as follows:project of an IoT system for air pollution monitoring,use of a data transmission security protocol,adaptation of MQTT and Amelia protocols for securing communication,automated and informal security analysis of the Amelia protocol adaptation.

## 2. Related Works

In literature, many IoT-based monitoring systems use micro-sensors for data gathering [9,10,11,12,13,14,15,16,17]. It should be mentioned that there are studies that propose a system for both internal and external air pollution, e.g., [9]. This work studies an IoT-based system implemented in a university area. The platform proposed in [9] offers individual course monitoring established on Wi-Fi tracking. Environmental monitoring refers to observing and measuring the environment to gather information about its condition and any changes that may occur. It involves using various tools and techniques to collect data on factors such as air quality, water quality, soil conditions, and weather patterns. The data collected through this process is used to identify potential environmental issues and to inform decisions regarding the management and protection of the natural environment. The system allows measuring multiple ecological parameters such as noise level, light, temperature, humidity, CO and $NO_{2}$ concentration. The monitoring system proposed in this paper is built based on the MiCS-4514 MicroElectroMechanicalSensor (MEMS) and popular hardware—Raspberry Pi and Arduino, which has a shallow level of safety data transmission.

Most studies focus only on one type of air pollution: indoor [10,11,12] or outdoor [13,14,15,16,17]. One of the examples of the former can be found in [10]. This research proposes a real-time monitoring procedure for gauging ambient air quality, which is low-cost, portable and based on IoT without transmission protection. The detection unit measures air grade elements such as $CO_{2}$, CO, $PM_{10}$, $NO_{2}$ using the sensors GP2Y1010AU, MH-Z14, MICS-4514 and DHT22. Another low-cost IoT system for real-time indoor air quality monitoring, without transmission protection, is proposed in [11]. This system uses the ESP8266 module as an analysing and transmission unit and the MICS-6814 detector as a sensing unit. Such a sensor allows for collecting data about a few air rate notes, such as CO, $NO_{2}$, $C_{3}H_{8}$ (propane), $C_{4}H_{10}$ (butane), $CH_{4}$ (methane), $H_{2}$ (hydrogen). Moreover, the system provides mobile software for constant announcements. Therefore, the user is alerted when the concentration of gases is exceeded. An attractive indoor air quality monitoring study is presented in [12]. The motivation for this research was that during the COVID-19 pandemic, people spent a lot of time in indoor environments. This paper proposes low-cost IoT sensor networks for indoor air quality assessment with standard unprotected data transmission. The presented system uses Plantower PMS5003 PM sensors deployed in four rooms (bedroom, living room, kitchen and office). Based on the data collected in the experiment, it was concluded that PM levels exceeded WHO 2021 annual ambient particulate concentration guidelines in all rooms.

An example of a system monitoring outdoor air pollution is proposed in [13]. In this work, the authors presented the innovation and implementation of an IoT detector instrument. Sensors adopted in this approach were both fixed and moving IoT sensor devices. To build the real-time air quality monitoring IoT sensing network, authors used i.a. Arduino Mega, Raspberry Pi, GPS Sensor, temperature and humidity sensor, microdust sensor and carbon dioxide sensor. It should be mentioned that this study adopted a machine learning technique to design an air quality prediction model. A prototype of a low-cost IoT system for Air Quality Index (AQI) monitoring is proposed in [14]. This paper used a Raspberry Pi module, a low-cost PM sensor, and some environmental detectors to measure diverse parameters such as temperature, humidity, and pressure.

Moreover, the same three-layered hierarchical distributed architecture for traffic flow monitoring was used. In the paper, [15], a Markov chain-based IoT approach is created to monitor, analyse and predict metropolitan air quality. The offered system is merged with an automobile to collect air quality data. As the microcontroller board, the authors used an Arduino Mini Pro. The sensing system operates an MQ2 gas sensor for counting CO concentration and a GP2Y1010AU0F visual detector for $PM_{10}$ and $PM_{2.5}$ concentrations. Another air quality monitoring system employing IoT methods for outdoor air pollution is proposed in [16]. The authors used a low-power vast space network to transmit the real-time data collected by portable particle matter, humidity and temperature sensors. The data gathered in the urban area confirms the correlation of the $PM_{2.5}$ concentration with many meteorological variables (e.g., the wind speed). An example of an IoT-based air pollution monitoring system can also be found in [17]. The system proposed by the authors employs multiple sensors (MQ135, MQ9 and MQ2) capable of detecting various air pollutants or gases. Using these sensors in conjunction with ESP Microntoroller, they have created a system to monitor the collected data through the smartphone Blynk application with a standard security level.

## 3. Proposed IoT System

### 3.1. System Structure

Air pollution can affect people’s life processes, deteriorate their health, and even cause death. In this case, air quality monitoring is a crucial part of daily life. The proposed system monitors air quality in some areas and alerts concerned locals to potential dangers. The system consists of four layers: data collection, end devices, central server and communication.

Only autonomous vehicles (AVs) operate in the data collection layer [18]. These vehicles are equipped with air pollution sensors and high-performance computing servers. AVs move around specific urban areas to collect and process air quality data. The air pollution sensors communicate with servers via GSM modules. The processed data is automatically made available to customers through messages containing statistical data and a map of a given area with marked measurement points of air pollutant concentrations. After finishing AV work, they return to the base and connect with the central server to transmit all collected data.

All IoT devices, like smartphones or tablets, work in the end devices layer. Users interested in air quality information can connect their devices and read shared information. Users should install a client application on their devices. The client application allows logging into the server and downloading air quality information. Figure 1 shows the air quality monitoring system’s main part structure that creates data collection and end device layers.

A central server works in the last system layer. This is also a high-performance computing server with different tasks than servers installed at AVs and with high memory resources. The central server manages all the work of autonomous vehicles through the server application. The first server’s task is defining routes where vehicles will move in a given area, and the server administrator can determine which streets will visit each car during the current route. The second server’s task is to collect and process all data about air quality. After finishing the route, AV must return to the base, connect to the central server, and send each collected information. Next, the central server will process each AV’s information, prepare statistics and charts, or determine dependencies. The third server’s task is to publish these statistics, charts or other summaries for the whole city on its website. Figure 1 summarises central server layer work.

The communication layer controls and secures two types of communication: between AV and devices and between the central server and AV. For the first communication type, we use the Message Queue Telemetry Transport (MQTT) protocol [19,20]. This is a simple, secure and dependable data transmission protocol. Further, it is suitable for devices that can only communicate at a certain transmission speed. The MQTT supports and secures users’ (devices’) registration and authentication processes.

Moreover, this protocol resists many attacks [20]. The MQTT base is the publish-and-subscribe model. The typical publish-and-subscribe model distinguishes three roles: Publisher, Subscriber and Broker. We assumed publisher, subscriber, main broker, and semi-broker roles in the proposed system.

The main broker role performs the central server. It shares information about air quality in the whole surveyed area and Semi-Brokers in the network. In Semi-Brokers’ case, the Main-Broker shares information about the area they are visiting. The autonomous vehicle is a Publisher and Semi-Broker. The server installed in AV allows the performing of these two roles. AV collects data about pollution in a specific urban area. Next, the server processes them and publishes them for the user. Users can connect to the AV’s server to download information. The server publishes information on a specific topic, such as air quality on some street, the concentration of the selected substance in the air, or a mix of such information. Thus, the end device is a subscriber. It subscribes to a specific Publisher’s topic or topics.

First, the user must connect to the Main-Broker to learn and subscribe to the appropriated Semi-Broker’s parameters. The user can connect to the nearest Semi-Broker or one of the Semi-Brokers in the network. In the first case, the Main-Broker will send them the parameters of the nearest Semi-Broker based on his geographical coordinates. In the second case, the user will choose one of the Semi-Brokers from the list in the application.

We adopted the Amelia protocol mentioned in [21] for the second communication type. The protocol’s default version protects against false links to Internet events. The first phase of this protocol provides a user registration process with a trusted Distribution Centre. The second phase of this protocol provides the validation process of a link to an event with a trusted Authentication Centre. Further, during this phase, users received symmetric keys for further communication.

We observed that the Amelia protocol is suitable for securing communication between AV and the central server, and we modified it. First, we combined a trusted Distribution Centre and Authentication Centre functionalities. The central server will generate necessary cryptographic objects, store information for communication and authenticate users (AVs, in the case of our solution). Hence, the scheme for the Amelia protocol’s registration phase in Alice–Bob notation will be as follows:
$\alpha_{1}$AV→CS:${\left\{i\left(AV\right),T_{AV}\right\}}_{K_{CS}^{+}}$$\alpha_{2}$CS→AV:$\{\#hash\left({\left\{UID_{AV},i\left(AV\right)\right\}}_{K_{AV}^{+}}\right),$

$T_{CS}^{AV}{\}}_{K_{AV}^{+}}$$\alpha_{3}$AV→CS:${\left\{T_{CS}^{AV}\right\}}_{K_{CS}^{+}}$

In this notation:${\left\{M\right\}}_{K}$ means the message *M* encrypted by key *K*,AV means the autonomous vehicle,CS means the central server,i(AV) is a AV’s text identifier,$T_{AV}$ is a AV’s timestamp,$K_{u}^{+}$ is the public key of *u* of the protocol participant, for example, the key $K_{CS}^{+}$ is the public key of the central server,$UID_{AV}$ stands for the numeric user identifier,$\#hash({\left\{UID_{AV},i\left(AV\right)\right\}}_{K_{AV}^{+}})$ means the execution of the hash function on the ciphertext ${\left\{UID_{AV},i\left(AV\right)\right\}}_{K_{AV}^{+}}$,$T_{CS}^{AV}$ means the timestamp generated by the central server for the unique user identifier AV.

In the first step of this protocol, AV tries to register to CS by sending its text identifier and freshly generated timestamp. Next, the central server checks its database to see if there is no such identifier. If everything is correct, it generates a numeric user identifier ($UID_{AV}$) and the timestamp $T_{CS}^{AV}$ and stores them with AV’s text identifier. In the second step, CS sends to AV hashed identifiers with the timestamp $T_{CS}^{AV}$. Note that, the $\#hash({\left\{UID_{AV},i\left(AV\right)\right\}}_{K_{AV}^{+}})$ value will be used to recognize AV by CS. Otherwise, it will stop communication. In the last step, AV confirms that it received the earlier message by sending the timestamp $T_{CS}^{AV}$ to CS.

The subsequent modifications we made were in the second phase of the Amelia protocol. In the protocol’s default version, the communication takes place between the user, the Authentication Center and the user who is the event organiser. In our system, AV will communicate only with the central server. We made the rest of the modifications in the message structures. Hence, the scheme of the authentication and communication phase of this protocol in Alice-Bob notation will be as follows:
$\alpha_{1}$AV→CS:$\{\#hash\left({\left\{UID_{AV},i\left(AV\right)\right\}}_{K_{AV}^{+}}\right),$

$T_{CS}^{AV}{,i\left(CS\right)\}}_{K_{CS}^{+}}$$\alpha_{2}$CS→AV:${\left\{T_{CS}^{AV},T_{CS},K_{CS-AV}\right\}}_{K_{AV}^{+}}$$\alpha_{3}$AV→CS:${\left\{T_{AV},DF_{AV}\right\}}_{K_{CS-AV}}$

In this notation:${\left\{M\right\}}_{K}$ means the message *M* encrypted by key *K*,AV means the autonomous vehicle,CS means the central server,i(u) is the text identifier of the user *u*,$K_{u}^{+}$ is the public key of *u* of the participant of the protocol; for example, the key $K_{AV}^{+}$ is the AV’s public key,$UID_{AV}$ stands for the numeric user identifier,$\#hash({\left\{UID_{AV},i\left(AV\right)\right\}}_{K_{AV}^{+}})$ means the execution of the hash function on the ciphertext ${\left\{UID_{AV},i\left(AV\right)\right\}}_{K_{AV}^{+}}$,$T_{CS}^{AV}$ means the timestamp generated by the central server for the unique user identifier AV,$DF_{AV}$ means a data file with information about air quality,$K_{CS-AV}$ is a symmetric session key shared between AV and CS.

In the first step, AV tries to authenticate to CS, sending its identifier with timestamp $T_{CS}^{AV}$ and CS’s identifier. The central server validates this data in the database. If the data are correct, it generates a symmetric session key shared between AV and CS. AV and CS will use this key later in the communication. Otherwise, it will stop communication. In the last step, the autonomous vehicle sends to the central server a data file with information about air quality with freshly generated timestamp $T_{AV}$. AV encrypts this message using the session key $K_{CS-AV}$.

The first phase must be executed once for each vehicle, but the second phase must be completed after every end of the route in the proposed system. Further, the proposed system uses secure communication channels via WiFi standards [22,23] for communication between the central server and AV, or LTE/5G standards mention [20,21] for communication between AV and devices.

Figure 2 shows the communication flow in the proposed system. We included here descriptions of each external and internal operation executed during the protocol’s steps and conditions for the subsequent steps executions. As mentioned, the MQTT network has a publisher, broker, and subscriber. The Publisher (autonomous vehicle in our solution, AV) must register on the Broker (central server, CS). In this case, it executes the first phase of the Amelia protocol. If the AV is registered correctly on CS, it can authenticate to CS, establish a session, and send data about air quality. Hence, it executes the second phase of the Amelia protocol. The sending of data to CS is equivalent to the publishing step performed by the publisher in the MQTT protocol. The Subscriber (user and his device) can subscribe to the published data. The Broker updates and publishes data for Subscribers regularly.

We verified the adaptation of the Amelia protocol using a tool for automated security protocol verification mentioned in [24]. This same tool was used to verify the Amelia protocol’s full version. Thanks to this tool, we can consider different executions of security protocols, including various time parameters such as time of composing the message, encryption/decryption times, or delay in the network, and also different network conditions (with or without attacker). We performed analysis using a computer unit with the Linux Ubuntu operating system, Intel Core i7 processor, and 16 GB RAM. Further, we used various values of time parameters during tests. The tool uses an abstract time unit [tu], which means some period.

Thanks to the tool mentioned, we could execute two types of research: timed analysis and simulations of the delays in the network. During one part of the research, we assumed the following values of time parameters: mechanism and one-way hash functions eliminate impersonation attacks.

time of generating confidential information: $T_{g}=1\left[tu\right]$,time of composing the message: $T_{c}=1\left[tu\right]$,asymmetric encryption or decryption time: $T_{e}=3\left[tu\right]$,symmetric encryption or decryption time: $T_{e}=2\left[tu\right]$,minimal delay in the network: $D_{min}=1\left[tu\right]$,maximal delay in the network: $D_{max}=3\left[tu\right]$,execution time of the hash function: H=2[tu].

For the timed analysis, the tool calculated the values of timed parameters summarised in Table 1. We observed that for each protocol part, four executions were possible: one execution with honest users and three executions with an attacker. Only executions with real users and executions with an attacker (like himself) were likely to be performed. They were executed while maintaining the imposed time conditions. Further, we analysed the messaging and learning flow in executions with the attacker who tries to impersonate other users. These executions did not meet time conditions. If the attacker wants to execute a subsequent step, he tries to establish the next session with CS and acquire appropriate knowledge. The attacker cannot obtain it because he cannot gain the necessary cryptograms or cryptographic objects to prepare messages according to protocol structure.

Next, we execute simulations of this protocol. This type of research examines the influence of delay in the network on execution time. Additionally, it demonstrates the capabilities of the attacker in a time-constrained setting. It is feasible to generate network delay values randomly based on four different probability distributions. We examined this protocol using all probability distributions implemented in the tool. This research confirmed the results of the timed analysis. Only executions with honest users and executions with an attacker (like himself) were possible to perform. The rest of the executions did not meet time conditions because of the attacker’s lack of knowledge.

Subsequently, an informal security analysis was conducted to validate our protocol’s attainment of the primary security features. We analysed such properties as anonymity and mutual authentication. Anonymity is achieved through asymmetric cryptography during the registration and authentication processes, which safeguards the user’s identity. In the context of mutual authentication, both the AV and the CS must possess a shared symmetric key to establish mutual authentication. The CS generates the key during the second authentication and communication phase step. AV and CS verify their identity during this stage by transmitting their shared timestamps and identifiers.

Additionally, we conducted an ad hoc security study to ascertain potential vulnerabilities that could compromise the effectiveness of our protocol. We analysed the aforementioned offensive scenarios. Adapting the Amelia protocol effectively mitigates replay attacks through a timestamp mechanism. The initial step involves the verification of the timestamp by each recipient. The continuation of contact is contingent upon receiving the message within the prescribed limit; otherwise, the recipient will terminate the communication. Further, this protocol mitigates the risk of a man-in-the-middle attack by incorporating a session key, which is shared among the parties involved. The encryption key employed in this context renders it impossible for the attacker to extract any constituent elements from the encrypted messages. Suppose an attacker successfully intercepts a message and attempts to transmit it during a subsequent session. In that case, the recipient will decline acceptance of the message after verifying the timestamp. Using symmetric and asymmetric encryption, timestamp mechanism, and one-way hash functions eliminates impersonation attacks.

### 3.2. Vehicle

During research, the critical role is to ensure the correctness of the data. Wrong data give wrong results. Therefore, any interference should be eliminated during measurements. For this reason, we used a Nissan Leaf electric car for data collection in our experiment (see Figure 3). Its most significant advantage in this case is that it does not emit air pollution during testing.

### 3.3. Instruments

The measuring device consists of two modules built into one Peli-type case:a measuring module with a built-in suspended dust measurement system,a measuring module with gas sensors working in a pump system.

Technical specification of the measuring instrument:$PM_{10}$ dust range from 0 to 1999.9 μg/m^3^, sensitivity 0.3 μg/m^3^$PM_{2.5}$ dust range from 0 to 999.9 μg/m^3^VOC Volatile Organic Compounds PID 10.6 eV detection range: 0 ppm to 50 ppm, photoionization method,CO range from 1 to 500 ppm, the electrochemical method$NO_{2}$ range from 0 to 20 ppm, the electrochemical methodNO range 0 to 20 ppm, the electrochemical method$SO_{2}$ range from 0 to 50 ppm, the electrochemical methodtemperature −40 °C to 85 °C, resolution 0.1 °Chumidity from 0 to 100pressure from 300 hPa to 1100 hPa, accuracy 100 PaGPS position (GNSS) accuracy up to 1.5 mGSM/3G/LTE moduleWiFi 2.4/5.8 GHz 802.11an.

The integrated measuring instruments have been placed in a carrying case. Such a research set has been equipped with magnetic holders to place it on the car’s roof. Thanks to such placement, the measurement of air pollution takes place at a height comparable to the height of the human nose. This solution ensures the reliability of measurements and allows us to draw valid conclusions regarding the level of air pollution and the impact of individual gases and dust on human health. The power supply for the measuring set (grey cable in Figure 4) came from the cigarette lighter socket. The set had a GPS module (black cable in the Figure 4).

### 3.4. Results

The Table 2 and Table 3 present data collected by the pollution measurement system discussed in the article. Measurements were taken on two different days. Each row corresponds to information gathered at a specific time (first column) and location (second, third, and fourth columns). The subsequent columns contain data regarding harmful gases and particulate matter concentration in each location. The last columns provide information about temperature and humidity, which can influence the intensity of perception of pollution.

Upon analysing the obtained results, we concluded that atmospheric pressure affects the concentration of suspended particles $PM_{10}$ and $PM_{2.5}$, although its influence is not direct. Atmospheric pressure can impact weather conditions, which, in turn, may affect the concentration of these pollutants. The primary ways atmospheric pressure may influence $PM_{10}$ and $PM_{2.5}$ concentrations include weather cycles or thermal inversions.

Regarding weather cycles, changes in atmospheric pressure are typically associated with various weather conditions. For instance, high atmospheric pressure often favours calmer weather with reduced wind, potentially leading to increased accumulation of $PM_{10}$ and $PM_{2.5}$ particles in the atmosphere.

On the other hand, concerning thermal inversions, atmospheric pressure may influence their occurrence. Under conditions of high atmospheric pressure, thermal inversions might be more likely. Thermal inversions can trap pollutants in the lower atmospheric layers, leading to increased concentrations of suspended particles.

Additionally, atmospheric pressure changes often coincide with humidity changes, corroborated by the obtained measurement data. High humidity may promote the accumulation of suspended particles, potentially resulting in increased concentrations of $PM_{10}$ and $PM_{2.5}$.

It is worth noting that the influence of atmospheric pressure on the concentration of suspended particles is often indirect and related to weather conditions resulting from these pressure changes. However, atmospheric pressure alone does not directly cause the increase or decrease in $PM_{10}$ and $PM_{2.5}$ concentrations.

The data obtained by the system can subsequently be made available to subscribers of the service offered by the system. There are two possibilities for the graphical presentation of measurement results. The first option is to present data in a graphic form (Figure 5), where the circle size shows the degree of pollution, and the red colour informs about exceeding the standard concentration of gases or dust. This circle is the so-called heat map that enables data visualisation that shows the size of individual values in a data set as a colour. Colour differences may be due to shade or intensity. A more intense colour (red or close to red) indicates increased pollutant concentration in the air. The heat map considers the values of all pollutants for a specific geographic point. The second way to show the results is to display numerical values (Figure 6) in a given location, and standards violations are translated into a change in the font colour. In this way, people sensitive to dust can avoid moving in the area. Meanwhile, people with poor lung ventilation due to harmful gases may be informed of the danger. Monitoring air quality in city centres using professional mobile sensors offers real-time insights into pollution levels. These sensors provide accurate data, enabling prompt actions to mitigate health risks associated with poor air quality. Improved monitoring aids in identifying pollution sources, allowing authorities to take targeted measures for cleaner air. Deploying such sensors strategically across urban areas creates a comprehensive network for continuous assessment. Accessible data from these sensors empowers citizens to make informed decisions about outdoor activities and health precautions. Real-time alerts and data dissemination to users via apps or online platforms promote awareness and proactive measures. Mobile sensor networks facilitate community engagement, fostering collaborative efforts to address air quality concerns. Regular monitoring enhances policy-making by providing evidence-based information for effective regulations. The portability of these sensors allows for flexibility in relocation, catering to dynamic changes in pollution hotspots. Employing professional mobile sensors and distributing their data to users amplifies public awareness and drives collective actions towards cleaner, healthier urban environments.

## 4. Conclusions

This article highlights the issue of air pollution, especially outdoors, and its severe impact on human health. IoT-based air quality monitoring systems are proposed as a practical solution to monitor and publish real-time air quality information. Several examples of IoT-based air quality monitoring systems using micro-sensors for data collection are presented for indoor and outdoor environments. The systems use various sensors to measure air quality parameters such as $CO_{2}$, CO, $PM_{10}$, $NO_{2}$, temperature, and humidity. The article emphasises the need for effective air quality monitoring and the potential of IoT-based systems to address this issue. An important issue raised in this work is the security of data transmission. A data transmission protocol has been proposed to ensure security and resistance to attacks related to the interception of transmissions in a public area.

The main limitation of the system is its autonomy. Currently, fully safe and reliable autonomous vehicles have not been developed. Such a solution would allow the entire process to be automated. This will become possible when effective algorithms for vehicle autonomy are developed.

Future work will focus on developing a user-friendly interface for multiple software platforms.

## Figures and Tables

**Figure 1 sensors-24-00445-f001:**
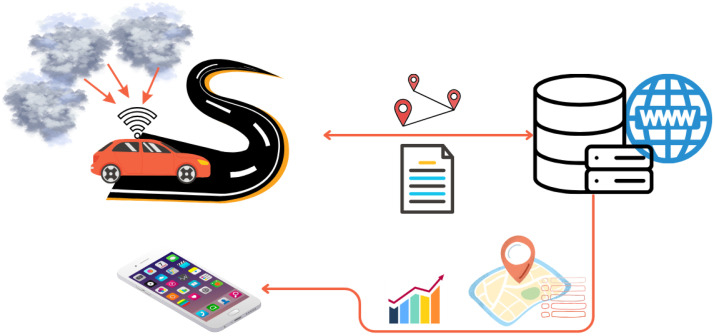
Structure of air quality monitoring system main part.

**Figure 2 sensors-24-00445-f002:**
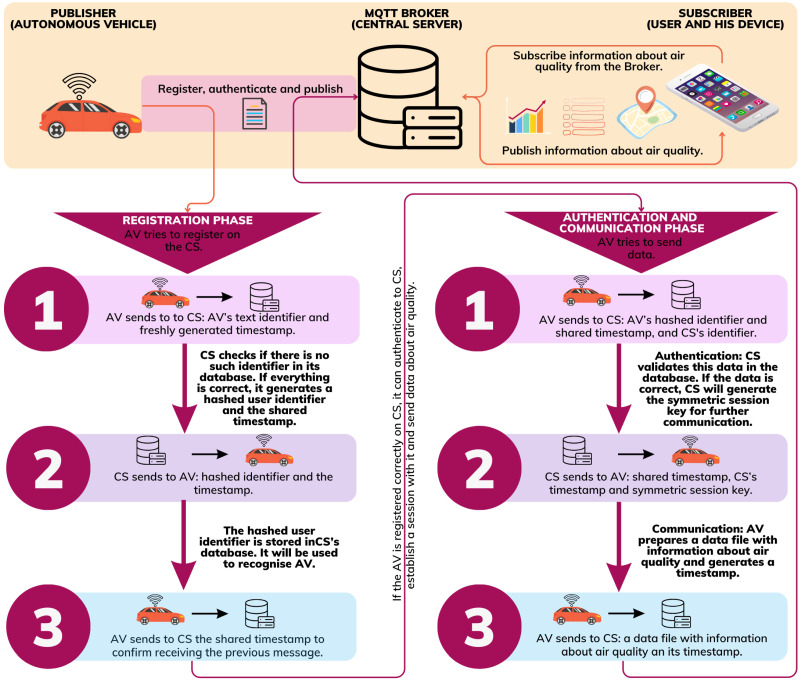
Communication flow in the proposed system.

**Figure 3 sensors-24-00445-f003:**
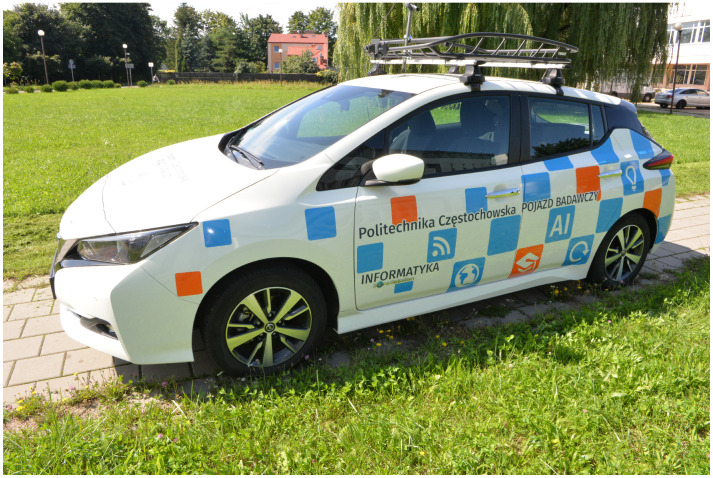
Vehicle used in experiment.

**Figure 4 sensors-24-00445-f004:**
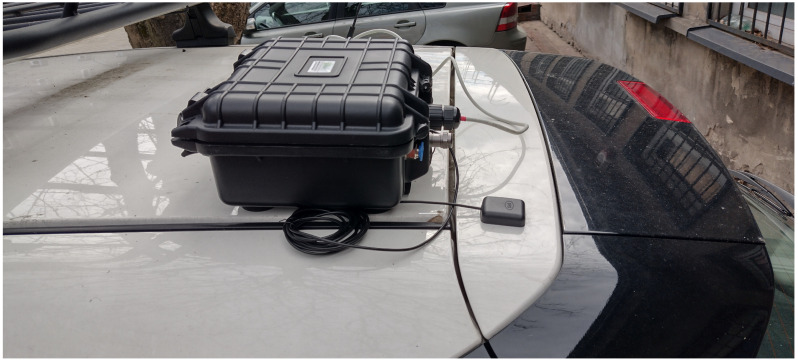
Instrument used in experiment.

**Figure 5 sensors-24-00445-f005:**
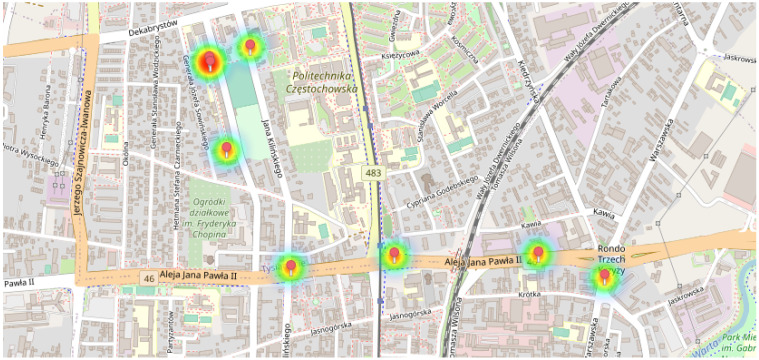
Results of measurement in the form of maps with heatmaps.

**Figure 6 sensors-24-00445-f006:**
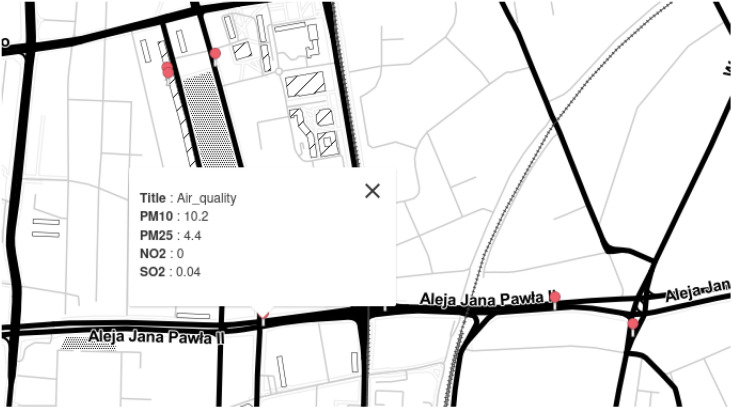
Results of measurement in the form of maps and numbers.

**Table 1 sensors-24-00445-t001:** Timed parameters used during research.

Parameter	Registration Phase	Authentiction and Communication Phase
Minimal $\alpha_{1}$ step time	9 [tu]	11 [tu]
Minimal $\alpha_{2}$ step time	14 [tu]	16 [tu]
Minimal $\alpha_{3}$ step time	8 [tu]	10 [tu]
Maximal $\alpha_{1}$ step time	8 [tu]	10 [tu]
Maximal $\alpha_{2}$ step time	10 [tu]	12 [tu]
Maximal $\alpha_{3}$ step time	7 [tu]	9 [tu]
Minimal session time	31 [tu]	25 [tu]
Maximal session time	37 [tu]	31 [tu]
Timeout for $\alpha_{1}$ step	10 [tu]	9 [tu]
Timeout for $\alpha_{2}$ step	26 [tu]	21 [tu]
Timeout for $\alpha_{3}$ step	37 [tu]	31 [tu]

**Table 2 sensors-24-00445-t002:** The data of air quality—day 1.

Time	LAT	LON	ALT	${\mathrm{PM}}_{10}$	${\mathrm{PM}}_{2.5}$	NO_2_	SO_2_	NO	PID	Temp	Press	Hum
	[°]	[°]	[m]	[μg/m^3^]	[μg/m^3^]	[ppm]	[ppm]	[ppm]	[ppm]	[°C]	[hPa]	[%]
14.11.2022 08:46	51.253555	22.563112	0	9.2	5.4	0.129	0	0.097	0.199	8.88	1007.26	68.89
14.11.2022 08:45	51.253555	22.563112	0	9.2	5.4	0.129	0	0.097	0.199	8.88	1007.26	68.89
14.11.2022 08:44	51.253555	22.563112	179.6	8.1	4.3	0.12	0	0.11	0.195	8.76	1007.06	68.9
14.11.2022 08:43	51.253135	22.560657	183.2	6.9	3.7	0.136	0	0.223	0.198	9.04	1007.17	68.27
14.11.2022 08:41	51.251875	22.551793	191.4	9.9	4.8	0.078	0	0.123	0.187	9.51	1006.38	66.89
14.11.2022 08:40	51.250048	22.552458	198.5	7.6	4.4	0.142	0	0.171	0.182	9.63	1004.89	65.52
14.11.2022 08:39	51.249043	22.554722	192.2	11.2	4.9	0.249	0	0.858	0.207	9.68	1005.28	65.99
14.11.2022 08:38	51.249638	22.55818	199.4	8.6	4.6	0.116	0	0.284	0.183	9.82	1004.94	65.06
14.11.2022 08:37	51.251273	22.558928	182.1	9	4.8	0.097	0	0.353	0.181	9.94	1006.06	64.2
14.11.2022 08:36	51.253255	22.561388	172.2	9.1	4.7	0.071	0	0.106	0.173	10.21	1007.28	62.32
14.11.2022 08:35	51.253255	22.561388	172.2	19.5	10.9	0.076	0	0.259	0.176	10.61	1007.33	61.47
14.11.2022 08:34	51.253093	22.563277	171.9	7.9	4.7	0.157	0	0.12	0.172	11.1	1007.52	58.85
14.11.2022 08:33	51.253477	22.56299	177.8	7.1	3.8	0.073	0	0.101	0.17	11.86	1007.43	55.57

**Table 3 sensors-24-00445-t003:** The results of measurements—day 2.

Time	LAT	LON	ALT	PM_10_	${\mathrm{PM}}_{2.5}$	NO_2_	SO_2_	NO	PID	Temp	Press	Hum
	[°]	[°]	[m]	[μg/m^3^]	[μg/m^3^]	[ppm]	[ppm]	[ppm]	[ppm]	[°C]	[hPa]	[%]
2023-05-08 14:00	50.823513	19.11048	240.1	4.8	1.9	0	0.067	0.25	0.175	19.96	998.15	24.68
2023-05-08 13:59	50.823378	19.110498	243.6	5.9	1.7	0	0.063	0.263	0.177	19.3	998.13	25.94
2023-05-08 13:58	50.820803	19.111305	254.5	5.6	2.2	0	0.043	0.29	0.172	18.86	998.68	24.56
2023-05-08 13:56	50.823888	19.112418	315.1	4.5	1.8	0	0.058	0.319	0.179	20.78	998.37	22.78
2023-05-08 13:55	50.817292	19.114335	258.9	10.2	4.4	0	0.04	0.606	0.186	21.38	999.04	22.15
2023-05-08 13:54	50.817657	19.119252	263	11.1	2.8	0	0.027	0.399	0.185	22.84	999.26	20.24
2023-05-08 13:53	50.817695	19.12604	264.9	5.6	2.2	0	0.077	0.395	0.189	24.51	999.19	19.46
2023-05-08 13:52	50.817007	19.1292	260.0	7.8	2	0	0.076	0.332	0.188	24.52	999.57	18.93
2023-05-08 13:51	50.811723	19.130872	255.1	6.3	2.3	0	0.054	0.98	0.196	25.28	999.51	18.29
2023-05-08 13:50	50.811863	19.129823	254.8	5.4	2.1	0	0.069	0.324	0.191	25.42	999.44	17.82
2023-05-08 13:48	50.811735	19.123877	272.7	5.7	2.1	0	0.086	0.452	0.203	26.69	998.62	17.21
2023-05-08 13:47	50.811838	19.12062	264.3	8.5	2.8	0	0.07	0.409	0.205	26.85	998.57	17.18
2023-05-08 13:37	50.811457	19.113898	263.5	4.9	2.2	0	0.117	0.456	0.24	19.33	998.48	24.41
2023-05-08 13:31	50.806778	19.103398	273.5	5.5	2.3	0	0.233	0.69	0.305	16.54	997.53	29.27
2023-05-08 13:30	50.805333	19.103503	271.7	7.9	3	0	0.253	0.662	0.319	16.56	998.03	28.75
2023-05-08 13:29	50.806027	19.103187	270.9	7.1	2.3	0	0.3	0.882	0.337	17.09	997.9	28.29
2023-05-08 13:28	50.80842	19.103825	270.2	6.1	2.2	0	0.355	0.998	0.368	17.37	997.4	28.09
2023-05-08 13:27	50.810742	19.104067	268.0	6.3	2.1	0	0.404	0.729	0.37	17.69	997.61	26.86
2023-05-08 13:26	50.812808	19.104128	266.6	5.4	1.8	0	0.509	1.033	0.405	18.42	997.45	25.82
2023-05-08 13:25	50.817035	19.105248	267.4	5.3	1.7	0	0.687	0.881	0.44	19.13	998.57	25.4
2023-05-08 13:24	50.81727	19.112653	269.5	4.8	1.4	0	0.992	1.118	0.459	20.35	998.96	23.28
2023-05-08 13:23	50.820793	19.11131	282.5	4.8	1.7	0	2.172	1.68	0.509	22.02	998.56	22.18
2023-05-08 13:22	50.822162	19.110895	275.6	5.9	2.1	0	14.246	3.834	0.63	23.52	998.31	23.01

## Data Availability

Data are contained within the article.

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
