# Peer review of "An IoT System for Air Pollution Monitoring with Safe Data Transmission"

_sensors, 2024, doi:10.3390/s24020445_

Round 1

Reviewer 1 Report

Comments and Suggestions for Authors

Dear authors, 

your work is really up-to-date, you refer to many contributions to meetings. I think this is fantastic. However, I missed your reference #20. I think it is incomplete.

I have found, too, a couple of typing errors:

Line #230: (see Fig.3)see Figure 3)

Line #237 and line #238 both are about PM10 sensors? I think one of them must be PM2.5 sensor.

Line #257: (grey cable 4 in fig 4) I think there are too many 4 in this sentence. 

Figure 4 is a good one, but figure 5 is unnecessary.

I have a question about the car: when the car is measuring, is it running? Yes, you say in line #231 that it is an electric car, so it does not emit any gases, but when it is running, its tyres really do emit particulates.

I have a question about table 1: in what units do you show results? 

I have a question about figure 6: You say that the size of the circle indicates the degree of pollution. What does the colour mean? How do you know which pollutant you are talking about?

I have a question about figure 7: It says air_quality, but it shows pollutant concentrations, doesn't it? What units are you using?

In fact, people who read these numbers on their phones do not really know whether these numbers mean risk or not. It can be a bit confusing for people.

Author Response

Responses to Reviewer 1:

1) Dear authors,

your work is really up-to-date, you refer to many contributions to meetings. I think this is fantastic. However, I missed your reference #20. I think it is incomplete.

Authors' response: We wish to thank the reviewer for her/his careful review and her/his comments and suggested improvements to the paper. We corrected reference 20.

2) I have found, too, a couple of typing errors:

Line #230: (see Fig.3)see Figure 3)

Authors' response: Thank you for this remark. We corrected this error.

3) Line #237 and line #238 both are about PM10 sensors? I think one of them must be PM25 sensor. 

Authors' response: Yes, corrected.

4) Line #257: (grey cable 4 in fig 4) I think there are too many 4 in this sentence.

Authors' response: Thank you for this remark. We corrected this error.

5) Figure 4 is a good one, but figure 5 is unnecessary.

Authors' response: Yes, we removed the photo.

6) I have a question about the car: when the car is measuring, is it running? Yes, you say in line #231 that it is an electric car, so it does not emit any gases, but when it is running, its tyres really do emit particulates.

Authors' response: Correct. It was not possible to remove all sources of potential interference. I think that at speeds of 30 km/h the interference is negligible, especially since the sensors are located on the car's roof.

7) I have a question about table 1: in what units do you show results?

Authors' response: The units have been replenished.

8) I have a question about figure 6: You say that the size of the circle indicates the degree of pollution. What does the colour mean? How do you know which pollutant you are talking about?

Authors' response: Thank you for these questions. These circles are the so-called heat map that enables data visualisation that shows the size of individual values in a data set as a colour. Colour differences may be due to shade or intensity. A more intense colour (red or close to red) indicates an increased level of pollutant concentration in the air. The heat map considers the values of all pollutants for a specific geographic point. We added an appropriate explanation in the text.

9) I have a question about figure 7: It says air_quality, but it shows pollutant concentrations, doesn't it? What units are you using? 

Authors' response: Thank you for these questions. The name "air_quality" indicated the name of the spreadsheet with data. The units are the same as in the Table 1.

10) In fact, people who read these numbers on their phones do not really know whether these numbers mean risk or not. It can be a bit confusing for people.

Authors' response: Thank you for this remark. We will add appropriate notifications in the user application.

Reviewer 2 Report

Comments and Suggestions for Authors

The authors proposed an IoT system for air pollution monitoring with safe data transmission.

The authors are advised to provide details about the proposed technique and the advantages of the proposed technique in the abstract.

Motivation and contribution are vital parameters; they should be given in the abstract. So the authors asked to add motivation and contribution in Section 1.

The authors discussed some IoT-based air quality monitoring systems in related work. But the outcome of related work is missing. The authors should highlight the limitations of existing work and show how the proposed system solves these limitations. Since the major motivation of the proposed work is safe data transmission, authors should highlight the data transmission process of existing work.

The authors used the Message Queue Telemetry Transport (MQTT) protocol and modified the Amelia protocol for secure data transmission. Please provide a flowchart or process flow diagram and explain it.

How was the performance of the proposed system evaluated? Please show how the system provides safe data transmission. Please prove how system resistance to attacks is related to the interception of transmissions.

Please give the limitations of the proposed work and the future scope of the proposed work.

Comments on the Quality of English Language

Please avoid the word “you” in the article. There are a few typographical and grammatical errors in the article; please clear up all those mistakes.

Author Response

Responses to Reviewer 2:

1) The authors proposed an IoT system for air pollution monitoring with safe data transmission.

Authors' response: We wish to thank the reviewer for her/his careful review and her/his comments and suggested improvements to the paper.

2) The authors are advised to provide details about the proposed technique and the advantages of the proposed technique in the abstract.Authors' response: Information on this has been added.

3) Motivation and contribution are vital parameters; they should be given in the abstract. So the authors asked to add motivation and contribution in Section 1.

Authors' response: Information on this has been added.

4) The authors discussed some IoT-based air quality monitoring systems in related work. But the outcome of related work is missing. The authors should highlight the limitations of existing work and show how the proposed system solves these limitations. Since the major motivation of the proposed work is safe data transmission, authors should highlight the data transmission process of existing work.

Authors' response: Information on this has been added.

5) The authors used the Message Queue Telemetry Transport (MQTT) protocol and modified the Amelia protocol for secure data transmission. Please provide a flowchart or process flow diagram and explain it.

Authors' response: Thank you for this remark. We added the figure with the appropriate description for the communication flow in the proposed system. Also, we combine two earlier Figures into one. 

6) How was the performance of the proposed system evaluated? Please show how the system provides safe data transmission. Please prove how system resistance to attacks is related to the interception of transmissions.

Authors' response: Thank you for these questions. We performed and presented automated and informal verification of the Amelia protocol adaptation.

7) Please give the limitations of the proposed work and the future scope of the proposed work.

Authors' response: As part of further work, we plan to verify the protocol in real conditions. During this research, we will obtain information about the limitations of the proposed solution.

Round 2

Reviewer 2 Report

Comments and Suggestions for Authors

The authors provided a limited amount of proposed technique details in the abstract. It would be good if authors provided more details in the abstract.

In this paper, autonomous vehicles are used for data collection (outdoor air quality monitoring system). So authors are requested to provide details about outdoor air pollution in the introduction.

The limitations and future scope of the proposed system should be added to the conclusion.

Please remove all grammatical and typographical mistakes from the paper.

Comments on the Quality of English Language

Please remove all grammatical and typographical mistakes from the paper.

Author Response

Answer for Reviewer

  1. In the abstract, detailed information about the system was added. (blue colour)
  2. It was marked with information about outdoor pollution. (green colour)
  3. In the conclusion, information about the limitations of the system and future work was added. (blue colour)
  4. Language correction was made.